# Theoretical and Experimental Study of the Effect of Plasma Characteristics on the Mechanical Properties of Ihram Cotton Fabric

**DOI:** 10.3390/membranes12090879

**Published:** 2022-09-12

**Authors:** Ahmed Rida Galaly, Nagia Dawood

**Affiliations:** 1Department of Engineering Science, Applied College, Umm Al-Qura University, Makkah 24381, Saudi Arabia; 2Department of Physics, Faculty of Science, Beni-Suef University, Beni-Suef 62521, Egypt; 3Physics Department, Faculty of Science, Taibah University, Al Madina Al Monawara 2363, Saudi Arabia

**Keywords:** radial distribution, electron temperature, weakly ionized plasma, cotton fabric, mechanical properties

## Abstract

Theoretical and experimental investigations of the radial distribution function of the electron temperature (RDFT), for the abnormal glow region in a low-density plasma fluid and weakly ionized argon gas, are provided. The final proved equation of RDFT agrees with the experimental data for different low pressures ranging from 0.2 to 1.2 torr, confirming that the electron temperatures decrease with an increasing product of radial distance (R) and gas pressures (P). A comparison of the two configurations: R>L and L>R,  for the axial distance (L), from the tip of the single probe to the cathode electrode, and the cathode electrode radius (R), shows that, in both cases, the generated plasma temperatures decrease, and densities increase. The RDFT accurately depicts a dramatic decrease for L < R by 60% compared with the values for L > R. This indicates that, when L < R, the rate of plasma loss by diffusion is reduced. Under this investigation, the mechanical characteristics of treated and pre-treated Ihram Cotton Fabric Samples were compared under the Influence of the different two configurations of Plasma Cell discharge: R>L>R. These characteristics included resiliency, strain hardening, tensile strength, elongation percentage, yield strength, ultimate tensile strength, toughness, and fracture (breaking) point. Furthermore, the mechanism parameters of plasma interaction with textile membrane will be discussed, such as: process mechanism, interaction, and gas type.

## 1. Introduction

The fourth state of matter, called plasma, is thought to be a relatively neutral medium (i.e., ion density equal to electron density) or it may be called quasi-neutralized. However, the characteristics of the plasma can be measured, such as: breakdown voltage, density, temperature, cathode fall thickness, negative glow thickness, positive column thickness, floating potential, etc. 

These characteristics can be measured using a diagnostic device like a probe (a small metallic electrode), where a very thin sheath forms surrounding the conducting surface, when the probe is put into a weakly ionized plasma as a result of the re-distribution of charges. Electron and ion motion close to the probe is governed by the amplitude and polarity of the probe potential. Only the ions can reach the probe surface when the probe potential is sufficiently negative. Thus, the probe current is equal to the ion random current (Iri) [1,2,3].

The most significant process occurring in the weakly ionized plasma is called ambipolar diffusion, and it is critical for the distribution of the plasma properties. The probe’s current–voltage characteristic curves can be used to calculate the plasma properties, such as electron temperature and electron density, under the assumption of a Maxwellian velocity distribution [4]. When there is a spatial gradient of the charged species, diffusion events take place in plasma [5,6]. The loss of neutrality [7] of the plasma is due to the faster diffusion of some charged species compared with others because the differently charged species have varying diffusivities.

However, if the Debye length is sufficiently small, a modest loss of neutrality also results in the induction of an ambipolar electric field, which slows down the fast-diffusing species and speeds up the slow diffusing species, maintaining the plasma’s quasi-neutrality [8].

Numerous studies have been performed on the experimental and numerical radial distribution of electron temperature and density from the plasma tube’s axis to its wall [9,10], but not on the theoretical derivation in terms of the impact of the Schottky condition and ambipolar diffusion because of the Bessel condition’s recurrence relation.

Nonthermal plasma has gained a broad interest and is used extensively for surface activation or modification, due to the ions, atoms, and molecules are relatively cold and do not cause any thermal damage to the materials surfaces [1]. Significant applications in the environmental control include sterilization of air, water, and surfaces [2]. Nonthermal plasma has many applications in a wide range of fields. There is a diverse range by an atmospheric pressure plasma jet such as: scavenging effect [3] of ascorbic acid and mannitol on hydroxyl radicals generated inside water; application of plasma in agriculture [4] for plant growth, use in the food industry; and against brain tumors [5] using preclinical cold atmospheric plasma cancer treatment.

In 2022, in the Kingdom of Saudi Arabia, especially in the holy city, Makkah, after the control of coronavirus 2019 (COVID-19) [11], from all over the world, millions of Muslim pilgrims returned to Hajj and Umrah. This necessitates many protections to reach a complete sanitization. One of the great challenges facing the authorities of Makkah is how to eliminate the contamination attached to the cotton fabric pieces of *Ihram* clothes from the surrounding environment. Muslim pilgrims must wear two pieces of cotton fabric (*Ihram* clothes) during the Hajj and Umrah seasons.

Because of its exceptional qualities, including regeneration, bio-degradation, softness, an affinity for skin, and hygroscopic properties, cotton fabrics are very well-liked. Cotton fabrics provide an excellent habitat for microbial growth when in touch with the human body because they can hold onto oxygen, moisture, and warmth as well as nutrients from spills and perspiration [12].

Many applications of plasma technology are being developed for the textile sector, as they can impart antibacterial properties, self-cleaning properties, UV resistance, antistatic properties, wettability, and dimensional stability to materials [13]. Plasma treatment of textiles is a new fashion technology, and research is growing due to it being an environmentally safe physical agent, especially in the antibacterial treatment of cotton fabrics [14].

In the present work, using the single probe as a diagnostic tool, a study of the radial dependence of the electron temperature in a low-density plasma was conducted. The results of the experiment were compared with a theoretical model for the radial distribution. Theoretical arguments are in agreement with experimental data on temperatures at various pressures in a low-pressure glow discharge of a DC (cold cathode) magnetron sputtering unit. Under the recent study of radial dependence of the electron temperature of Plasma Cell Discharges, one of the industrial applications, mechanical properties of ihram cotton fabric samples, will be studied, to prepare multifunctional properties for surface modification and to increase the performance quality of the cotton fabric.

## 2. Experimental Set-Up

### 2.1. System Preparations

Figure 1 displays a schematic diagram of the electrical circuit utilized in the author’s previous experiments to induce a DC glow discharge between two various plasma cell electrode configurations that occur inside of an evacuated stainless-steel chamber, with glass windows, to a pressure of 7 mTorr using a two-stage rotary pump [15,16]. Through a needle valve, high-purity gas was injected into the chamber. Using a 1200-volt DC power supply (Photon company, Cairo, Egypt), a stationary DC glow discharge was created between two electrodes of metallic disks at varied low Ar pressures and design changes. We used a Tektronix digital oscilloscope (Tektronix, Beaverton, OR, USA) to measure the applied voltage and discharge currents under the following conditions, ranging: from 0.5 to 5 Torr for applied pressures; from 100 to 1200 V for the discharge voltage; from 2 to 15 mA/cm^2^ for the current density, and from 4 to 90 mA for the discharge current rang. Two parallel circular copper electrodes, one of which serves as the plasma cell’s cathode and the other, the anode, which is placed at an axial gap distance of 5 cm, make up the plasma cell.

In order to prevent the accumulation of charged sheaths on the surfaces, confine, and intensify the plasma inside the discharge region between the two movable electrodes, the grounded holder for the Ihram cotton fabric samples is isolated from the stainless-steel outer chamber by polytetrafluoroethylene (PTFE) insulated material, where the axial distance (L) between the tip of the single probe and the cathode electrode, 5 cm and with 3 cm in radius (R), taking into account that we will deal with two conditions: R>L and L>R. As shown in Figure 1, a single spherical Langmuir probe, with the probe tip made from molybdenum wire (diameter 3.0 mm, length 0.5 mm), is inside the glow discharge plasma. To measure the axial potential distribution between the two electrodes, the single probe that is situated between the anode and the cathode, which is fixed at the ground potential, moves axially as discussed before in our previous work [17].

### 2.2. Textile Preparations

The tensile and elongation behaviors of Ihram cotton fabric samples, pre-treated and treated with the two different plasma reactor conditions R>L and L>R, were observed using the Zweigle Model Z010 (USTER technologies, Switzerland) in accordance with ASTM D412-98a under standardized atmospheric conditions and at a tension speed of 100 mm/min. Three measurements were taken, and the outcomes represented the average values. With a stable and uniform DC glow discharge, the mechanical characteristics of the pre-treated and treated samples were evaluated. The results were represented by the stress  σ (KPa) as a function of the strain  ε  (percent), where  σ=E ε, where *E* represents the Young’s modulus (stiffness) values.

The tensile and elongation behaviors of Ihram cotton fabric samples were pre-treated and treated with the two different plasma reactor conditions R>L and L>R. The measurements were made three times, and the results represented the mean values. The mechanical properties of the pre-treated and treated samples were tested with a uniform DC glow discharge, indicated by the stress  σ (KPa) as a function of the strain  ε  (percent), where  σ=E ε, with *E* representing Young’s modulus (stiffness) values.

## 3. Results and Discussion

### 3.1. The Radial Distribution of the Electron Temperatures

Inside the plasma fluid, the relation between the plasma charge density σ conductivity ρ, with *D_a_* being the ambipolar diffusion coefficient at the tube axis [18,19], is as follows:(1)$$D_{s}=D_{a}[1-\mu_{e}\;\;\frac{\rho }{\sigma }\;]$$
also taking into account that the flow of ions and electrons is the same, hence


***Γ*** = ***Γ******_i_*** = ***Γ******_e_***(2)


(Congruence approximation). If there is no external electric field, the fluxes of ions and electrons (drift–diffusion model) can be expressed as:


***Γ******_i_*** = −*D*_i_**∇***n*_i_ + *μ*_i_*n*_i_***E******_D_***(3)


***Γ******_e_*** = −*D*_e_**∇***n*_e_ + *μ*_e_*n*_e_***E******_D_***(4) where *E_D_* follows Poisson’s equation [20]. If we multiply the first equation by *μ*_e_*n*_e_ and e second one by *μ*_i_*n*_i_ and then subtract them, we obtain:


*μ*_i_*n*_i_*Γ*_e_ − *μ*_e_*n*e*Γ*_i_ = *μ*_e_*n*_e_*D*_i_**∇***n*_i_ − *μ*_e_*n*_e_*μ*_i_*n*_i_***E******_D_*** − *μ*_i_*n*_i_*D*_e_**∇***n*_e_ + *μ*_i_*n*_i_*μ*_e_*n*_e_***E******_D_***(5)


Since *Γ* = *Γ*_e_ = *Γ*_i_, with cancelling the terms of *E_D_*, and with using *(n_i_ = n_e_*), the flow of particles can then be written as follows:

Ambipolar diffusion coefficient is
(6)$$D_{a}=\;\frac{\;D_{e}\mu_{+}-D_{+}\mu_{e}\;}{\;\mu_{+}-\mu_{e}}$$
and *ρ* is the charge density in the plasma given by:(7)$$\rho =e[n_{+}-n_{e}]$$
where, at the tube axis, values of the conductivity *σ* are given by:(8)$$\sigma =e[\mu_{+}n_{+}-\mu_{e}n_{e}]$$

Consider that diffusion coefficient *D* is given by:(9)D=kTmν     m/s2
where *μ_ε_* are the mobility of electrons (*e*) and ions (+), *kT* (*ev*) is the particle temperature, and *ν* is the collision frequency between electrons and neutral atoms in Hz. By substituting with Equations (2)–(8) into Equation (1), we obtain
(10)$$D_{s}=\;\;\frac{\;D_{e}\mu_{+}-D_{+}\mu_{e}\;}{\;\mu_{+}-\mu_{e}}[1-\mu_{e}\frac{n_{+}-n_{e}}{\mu_{+}n_{+}-\mu_{e}n_{e}}\;]$$

From the Einstein relation, μ can be written as follows [21,22]:(11)D=⌈q⌉μ kT

By substituting into Equation (10) the value of μ, then:(12)$$D_{s}=\;\frac{\mu_{e}\;\mu_{+}\;KT{\;}_{e\;\;}-\mu_{e}\;\mu_{+}\;KT{\;}_{+\;}\;}{\left\lceil q\right\rceil \left(\;\mu_{+}-\mu_{e}\right)}[1-\mu_{e}\frac{n_{+}-n_{e}}{\mu_{+}n_{+}-\mu_{e}n_{e}}\;\;]$$
or
(13)$$D_{s}=\;\frac{\mu_{e}\;\mu_{+}\;\left(KT{\;}_{e\;\;}-KT{\;}_{+\;}\right)\;}{e\left(\;\mu_{+}-\mu_{e}\right)}[\frac{\mu_{+}n_{+}-\mu_{e}n_{e}-\mu_{e}n_{+}+\mu_{e}n_{e}}{\mu_{+}n_{+}-\mu_{e}n_{e}}\;]$$

Neglecting $kT{\;}_{+}$, where $kT{\;}_{e\;}\mathrm{is}\;\mathrm{much}\;\mathrm{greater}\;\mathrm{than}$ $\;kT{\;}_{+}$ finally results in

Then,
(14)$$D_{s}=\;\frac{n_{+}\mu_{e}\;\mu_{+}\;\left(\;kT{\;}_{e\;\;}\right)\;}{e\left(\;\mu_{+}\;-\;\mu_{e}\right)}\left[\frac{\mu_{+}-\mu_{e}}{\mu_{+}n_{+}-\mu_{e}n_{e}}\right]$$
$$D_{s}=\frac{n_{+}\mu_{e}\;\mu_{+}\;\left(\;kT{\;}_{e}\right)\;}{e}\left[\frac{1}{\mu_{+}n_{+}-\mu_{e}n_{e}}\right]$$

Then,
(15)$$D_{s}=\;\frac{\;\left(\;kT{\;}_{e\;\;}\right)\;}{e}\left[\frac{\mu_{e}\;\mu_{+}\;}{\mu_{+}-\mu_{e}\frac{n_{e}}{n_{+}}}\right]$$
or
(16)$$D_{s}=\;\frac{\;\left(kT{\;}_{e\;\;}\right)\;}{e}[\;\;\frac{\mu_{e}\;\;}{1-\frac{\mu_{e}}{\mu_{+}}\frac{n_{e}}{n_{+}}}]$$

Substituting μe=emeνe and μi=emiνi from Equations (9) and (11) into Equation (16) results in
(17)$$D_{S}=\frac{kT_{e}}{e}[\;\;\frac{\left(\frac{e}{m_{e}\nu_{e}{}^{}}\right)}{1-\;\;\;\;\;\;\;\;\;\;\frac{\left(\frac{e}{m_{e}\nu_{e}{}^{}}\right)n_{e}}{\left(\frac{e}{m_{i}\nu_{i}{}^{}}\right)n_{+}}\;}\;\;\;]$$
or
(18)$$D_{S}=kT_{e}[\frac{\left(\frac{1}{m_{e}\nu_{e}{}^{}}\right)}{1-\;\;\;\frac{m_{i}\nu_{i}{}^{}n_{e}}{m_{e}\nu_{e}{}^{}n_{+}}}]$$
or
(19)$$D_{S}=kT_{e}[\frac{n_{+}}{m_{e}\nu_{e}{}^{}n_{+}-m_{i}\nu_{i}{}^{}n_{e}}]$$
(20)$$\frac{D_{S}}{\nu_{i}}=\frac{kT_{e}}{m_{e}\nu_{e}{}^{}\nu_{i}{}^{}\;-\;\frac{m_{i}\nu_{i}{}^{2}n_{e}}{n_{+}}}$$

In experiments, the plasma production stage of the basil discharge is when the diffusion process, which is regarded as a crucial process, happens. Theoretically, the basil discharge process is connected to the recurrence relation for the Bessel condition, and the Schottky condition [23,24] may be proven using the following Section 3.1.1 and Section 3.1.2:

#### 3.1.1. Axial Distance Larger Than the Radial Distance (L>R)

R is radial distance from the center to the edge through the cathode electrode measured by the Langmuir probe moved radially, and L is the axial distance from the center of the cathode electrode far away vertically. Take into account the Schottky condition state that:(21)$$\frac{D_{S}}{\nu_{i}}={\left(\frac{R}{2.405}\right)}^{2}$$

By substituting Equation (21) into Equation (20), then
(22)$${\left(\frac{R}{2.4}\right)}^{2}=\frac{kT_{e}}{m_{e}\nu_{e}{}^{}\nu_{i}{}^{}\;-\;\frac{m_{i}\nu_{i}{}^{2}n_{e}}{n_{+}}}$$

Substituting the following equation into Equation (22)
(23)$$\nu_{e}=N_{n}\langle Q_{e-n}\upsilon_{e-n}\rangle$$
(24)νe=3.55×1016 PQe−n2kTeme12
(25)νi=3.55×1016 PQi−n2kTimi12
where *P* is the pressure in torr, $Q_{e-n}\;$ is the cross sections of electron-neutral and $\;Q_{i-n}\;$ is the ion-neutral collisions, respectively resulting in
(26)R2.42=kTe[me2kTeme122kTimi123.55×1016 P2  QiQe]−nen+ mi[3.55×1016 P2 Qi]22kTimi 

Then,
(27)RP2.42=14mekTimikTe123.55×10162QiQe−nen+ 3.55×1016Qi22kTimikTe

However, from Refs. [25,26],
$$\frac{\mu_{e}}{\mu_{i}}≅7.64\;{\left(m_{i}\right)}^{0.5}$$
or
(28)$$\;\;\;\;\;m_{i}≅{\left(\frac{1}{7.64}\frac{\mu_{e}}{\mu_{i}}\right)}^{2}$$

Then, substituting (11) into (28) gives
(29)1mi≅(7.64kTikTe)2

Moreover, substituting (29), $\frac{m_{i}}{m_{e}}≅$ 2000 and $\frac{\nu_{i}}{\nu_{e}}≅10^{-2}$ into (27) gives
(30)RP2.42=1[(7.64kTikTe)4mekTikTe123.55×10162QiQe]−nen+ 3.55×1016Qi2(7.64kTikTe)22kTikTe
(31)RP2=2.42[(7.64kTikTe)4mekTikTe123.55×10162QiQe]− nen+ 3.55×1016Qi2(7.64kTikTe)22kTikTe 

Substituting the values of $Q_{i},\;Q_{e}$, $\;m_{i},\;and\;m_{e}$ into (31) gives
(32)RP2=2.42  3.5527.642  (kTikTe)mekTikTe12]− nen+ 7.64kTikTe2kTikTe]
or
(33)RP2=2.42  3.5527.642kTikTe32[   me12− 7.64nen+   (kTikTe)32 ]

Then, finally,
(34)RP=0.17291kTikTe34[   me12− 7.64nen+   (kTikTe)32 ]12

#### 3.1.2. Axial Distance Less Than the Radial Distance (L<R)

The Schottky condition can be written as follows:(35)$$\frac{D_{S}}{\nu_{i}}={\left(\frac{R}{2.405}\right)}^{2}+{\left(\frac{L}{\pi }\right)}^{2}$$
and, by substituting Equation (20) into (35), then
(36)$${\left(\frac{R}{2.4}\right)}^{2}+{\left(\frac{L}{\pi }\right)}^{2}=\frac{kT_{e}}{m_{e}\nu_{e}{}^{}\nu_{i}{}^{}\;-\;\frac{m_{i}\nu_{i}{}^{2}n_{e}}{n_{+}}}$$

Substituting Equations (23)–(25) into Equation (36) respectively results in
(37)R2.42+Lπ2=kTe[me2kTeme122kTimi123.55×1016 P2  QiQe]−nen+ mi[3.55×1016 P2 Qi]22kTimi 

Then,
(38)(RP2.4)2+(Lπ)2=14mekTimikTe123.55×10162QiQe−nen+ 3.55×1016Qi22kTimikTe

Substitution from (28) and (29) into (38) gives
(39)RP2.42+Lπ2=1[(7.64kTikTe)4mekTikTe123.55×10162QiQe]−nen+ 3.55×1016Qi2(7.64kTikTe)22kTikTe
(40)RP2=2.42[(7.64kTikTe)4mekTikTe123.55×10162QiQe]− nen+ 3.55×1016Qi2(7.64kTikTe)22kTikTe+ −2.42 (Lπ)2 

Substituting the values of $Q_{i},\;Q_{e}$, $\;m_{i},\;and\;m_{e}$ into (40) finally gives
(41)RP=√(0.0299  kTikTe32[   me12− 7.64nen+   (kTikTe)32 ]−5.76 (Lπ)2)

The experimental study of radial dependence of the electron temperature in an argon discharge uses the data of the electron temperatures and densities from previous works for the abnormal cathode fall region (C.F.), using the semi-log curve of the electron current and the second derivatives of the electron current methods [Galaly2015], substituting values of *T_e_* of *N_e_* for the single spherical probe into Equations (34) and (41), for values of *T_e_* varying from 6.5 to 4.5 eV, and values of *N_e_* varied from (4 to 3) × 10^9^ cm^−3^.

Taking into account (*T_i_* = 0.1 *T_e_*), memi≅2000 [25,26,27], the ion density is given by n+=I+0.6  (kTem i)1/2 AP [28]. The area of spherical probe (*A_p_* = 4π r_p_^2^), with calculating positive ion current I_+_ from the I–V characteristic curve of the spherical single probe [29], *n*_+_ can be determined.

In the abnormal cathode fall region from the center to the edge of the electrode, Figure 2 compares the radial distribution of the electron temperature (RDFT) theoretically utilizing L > R for Equation (34) and L < R for Equation (41), depending on the presence of Ar as an inert gas and applied pressure of 1.2 torr. Theoretically, a dramatic radial decrement of RDFT in the electron temperature is found. Additionally, the edge effect causes the *T_e_* decreases at the electrode’s edge to be more pronounced than they are at the center [30]. The behavior for the radial distribution of *T_e_* for L < R accurately shows a dramatic decrease by 60% in comparison with L > R. This may be attributed to:
(a)*T_e_* decreases for increasing pressure may be due to [31]:(42)$$\left|Te\right|=\frac{13}{2}\;\;\left|P^{-66\times 10^{-3}}\right|$$(b)As the pressure increases lead to a further increase in the breakdown voltage, a sharp increase in electron density and a higher electron–electron collision frequency lead to the decrease of the electron temperatures [32].(c)Figure 2 shows that theoretical and experimental results are compatible and agree fairly well with von Engel assumptions [33].

**Figure 2 membranes-12-00879-f002:**
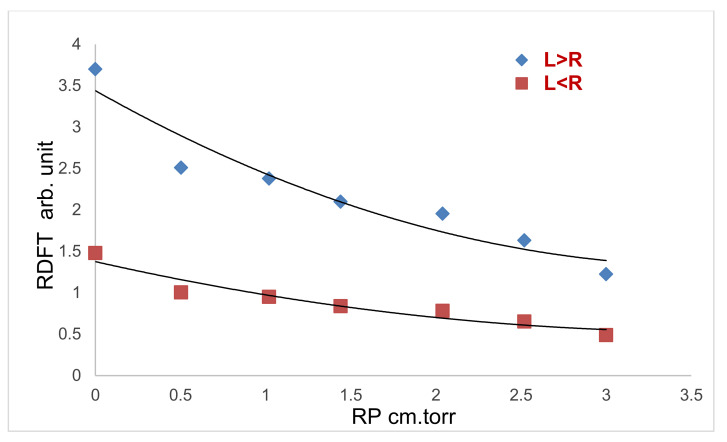
Theoretical and experimental comparison between the radial distribution of the electron temperature (RDFT) from the center (R*P* = 0) to the edge for R>L>R, at *P* = 1.2 torr.

### 3.2. Mechanical Properties

Figure 3 displays the tensile and elongation behaviors of the pre-treated and two-treated Ihram cotton fabric samples for the two different plasma discharges cases L < R and L > R, at *P* = 1.2 torr exposed to a uniform DC glow discharge of argon plasma, to test their mechanical properties as shown by the stress (KPa) as a function of the strain (percent). Additionally, Figure 3 depicts the linear region AB with straight lines denoted by σ=E ε, with the slope E denoting values for Young’s modulus (stiffness). The elastic region E  increased from 2.75 KPa for pre-treated samples to 3.2 KPa for samples treated with L > R, and to 3.5 KPa for samples treated with L < R [34].

The area under the curve of the elastic region AB represents the tensile resilience (R*T*) [35] as in:(43)$$(RT)=\frac{1}{2}\sigma \varepsilon .$$

R*T* indicates that the Ihram cotton fabric samples have a greater ability to absorb energy when elastically deformed for L > R and L < R, respectively, as shown in Equation (43): for pre-treated, treated with L > R, and treated with L < R, respectively, R*T* corresponds to values of 13,750, 16,000, and 17,500 J/m^3^:


(R*T*)_L__<R_ > (R*T*)_L__>R_ > (R*T*)_pre-treated_(44)


*WT* was calculated using the Microsoft Excel program by the area under the stress–strain curve from A up to the fracture (breaking point) D, representing the energy needed to extend the length of cotton fabric without damaging it and reflecting the mobility of the garment under deformation (up to fracture) [36,37]. *WT* increased as follows: 50,375 J/m^3^ for pre-treatment, 55,725 J/m^3^ for treatment with L > R, and 60,175 J/m^3^ for treatment with L < R, as shown in Equation (45):


(*WT*)_L__<R_ > (*WT*)_L__>R_ > (*WT*)_pre-treated_(45)


Table 1 compiles the mechanical characteristics of the pre-treated and treated samples and shows the following:(a)The L < R condition had a more favorable impact on the mechanical properties of plasma-treated Ihram cotton fabric samples than in the case L > R condition.(b)The elasticity area, stretch, and strain percentages of the Ihram cotton fabric samples were increased after treatment with plasma.

This can be attributed to the fact that

(a)The density and the energy of the positive ions emerging from the plasma cell for L > R were much greater than L < R, when colliding with the Ihram cotton fabric sample.(b)Higher potentials are anticipated since the cathode fall region is compressed in thickness, along with the negative glow and positive column regions when L < R. As a result, a powerful electric field is created, which causes the ions to accelerate and makes the sputtering process more effective [38].(c)For L < R, the generated plasma temperatures decrease and densities increase; this indicates that the rate of plasma loss by diffusion decreased in a manner similar to that of the applied magnetized DC plasma [39], leading to an increase in current and current density.

**Table 1 membranes-12-00879-t001:** The mechanical characteristics of the pre-treated and treated Ihram cotton fabric samples.

Position	Parameters	Units	Untreated	Treated with OMSE	Treated with OMSTE
From A to B	stiffness	KPa	2.75	3.2	3.5
B	yield	KPa	275	320	350
	strength				
	$\sigma_{Y}$				
C	ultimate	KPa	400	420	440
	tensile strength				
	$\sigma_{\mathrm{UTS}}$				
B–C	Strain	KPa	125	100	90
	Hardening				
	$\sigma_{\mathrm{UTS}}-\sigma_{Y}$				
D	Elongation percent at Breaking Point	%	200	230	250
Area under the curve of the elastic region	Resilience	J/m^3^	13,750	16,000	17,500
Area under the strain–stress curve up to fracture	Toughness	J/m^3^	50,375	55,725	60,175

### 3.3. The Mechanism Parameters of Plasma Interaction with Textile Membrane

#### 3.3.1. Process Mechanism

Electrons and ions are formed due to plasma discharge between the plasma cell electrodes. The Ihram cotton fabric sample on the holder becomes initially negatively charged relative to the plasma bulk, due to the higher mobility of the lighter electrons. Then, more electrons are repelled from the sample, and the positive ions are accelerated towards it [40]. Due to the scattering of the positive ions at the sample, chemical bonds are broken by energy transfer from reactive particles to the sample surface. Besides the chemical changes, there are physical changes due to the exposure of plasma. These changes produce more reactive surfaces and affect the mechanical properties more, producing either etching or dusty or damage for the sample.

#### 3.3.2. Interaction Type

The mechanical properties of the Ihram cotton fabric surface (with finished coded levels), which can be enhanced by the activation process, and modifications resulting from interactions between the plasma species and textile fibers, are the main determinants of the interaction mechanism between the plasma species and textile materials [41,42]. The covalent bonds on the surface of the Ihram cotton fabric sample are broken by the activation process, which also produces radicals. These are extremely reactive locations that mix with other species to form functional groups on the surface, such as organic molecules, unsaturated monomers, or reactive gases like oxygen [43,44].

Additionally, the etching procedure may be used with the activation procedure for L < R to clean the Ihram cotton fabric surface by bombarding the sample with ions, which eliminates impurities and pollutants [45]. Furthermore, the activation process was associated with dusty plasma because there was more scattering due to the greater separation between the two electrodes as in L > R case [46].

#### 3.3.3. Gas Type

When powerful ultraviolet photons and excited argon species, such as ions, electrons, neutrals, and meta-stables, attack the textile surface, they can break chemical bonds and start several reactions. Due to its high ablation efficiency and chemical inertness with the surface material, argon discharge can change the mechanical properties of textile surfaces [47]. Additionally, argon causes surface chain scissions (also known as activation) and crosslinking through reactions between inter- and intra-molecular polymer chains [48,49,50].

## 4. Conclusions

Experimental and theoretical investigations of the effects of the plasma parameters on the mechanical properties of Ihram Cotton Fabric were performed using the radial distribution function of the electron temperature for argon glow discharge, for the abnormal glow region in a low-density plasma and weakly ionized gas for both configurations R < L and R > L. The theory utilizes the cross sections of ion-neutral and electron-neutral collisions, the charge density of the plasma, conductivity at the tube’s axis, the Schottky condition, and the Einstein relation of ambipolar diffusion. For both configurations, R < L and R > L, respectively. It is possible to calculate the decrease in the radial distribution function of the electron temperature (RDFT) with increasing radial distance and gas pressure products. The theoretical investigation provides the radial decrement dependency of the electron temperatures from the center to the edge of the electrode and agrees well with experimental data of electron and ion temperatures. Experimentally, all the mechanical properties of the Ihram cotton fabric samples treated with plasma were found to be more positively influenced in the L < R condition than in the L > R condition. The use of plasma to treat the Ihram cotton fabric samples increased the elasticity area, the stretch, and the strain percentages. Our future work will involve an experimental study of the plasma treatment, not only of direct physical effects and mechanical changes but also of the antimicrobial property and quality caused by the plasma. The work will also involve conducting analytical investigations into the actual effect of the plasma treatment on the different cotton fabric samples.

## Figures and Tables

**Figure 1 membranes-12-00879-f001:**
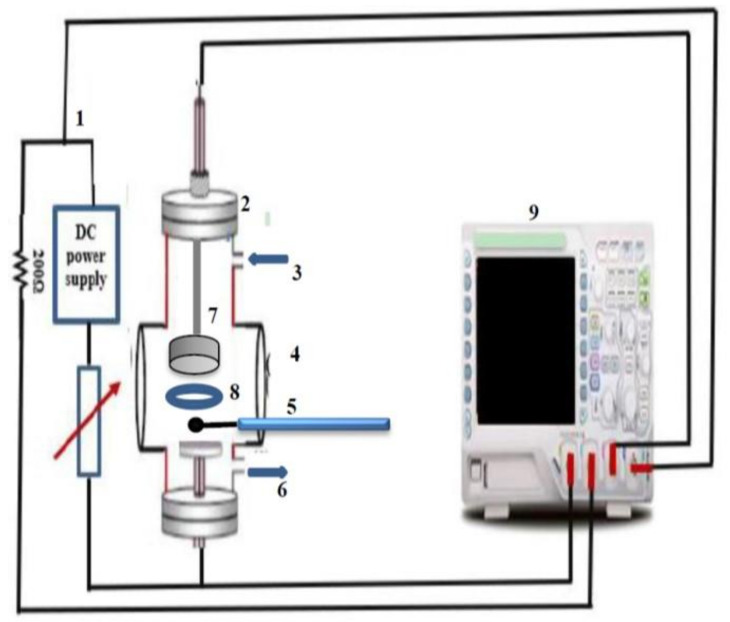
Schematic diagram of the experimental set-up of (**1**) DC glow discharge circuit, (**2**) evacuated chamber, (**3**) gas inlet, (**4**) window, (**5**) single probe, (**6**) rotary pump, (**7**) the plasma cell electrodes, (**8**) sample holder, and (**9**) oscilloscope.

**Figure 3 membranes-12-00879-f003:**
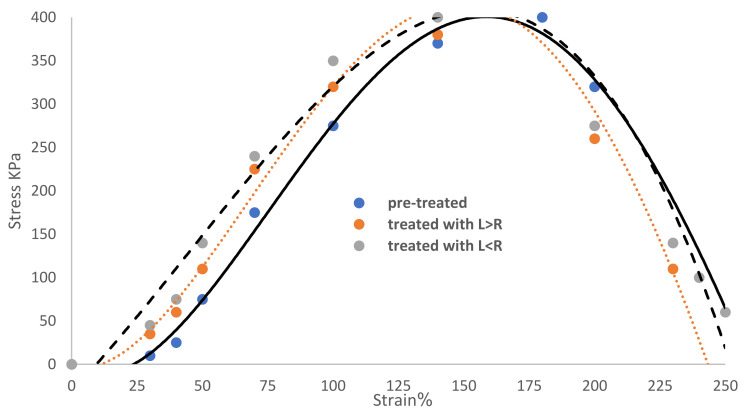
The measured mechanical properties of the pre-treated samples and treated samples with L < R and L > R.

## Data Availability

Data are contained within the article.

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
