# Peer review of "Theoretical and Experimental Study of the Effect of Plasma Characteristics on the Mechanical Properties of Ihram Cotton Fabric"

_membranes, 2022, doi:10.3390/membranes12090879_

Round 1
Reviewer 1 Report
Line 17:two configurations: L<R and L>R
69: Cotton fabrics provide
78: In the present work, using a single..
83: The RDFT of Plasma…applications, namely the improvement of the mechanical…
103: what is meant by “strengthen” the plasma ?
109: with the probe tip
170: demete ??
3.1.1 Axial..; (L>R)
192: 3.1.2 Axial distance smaller than the radial distance (L<R)
199: 204: Rewrite this paragraph (very difficult to understand) and give references for previous work.
225: Fig.2 indicate theoretical and experimental results in the caption
Fig. 3: improve quality of figure
Author Response
answers of comments

Reviewer 2 Report
Review on Theoretical and experimental study of the effect of plasma characteristics on the mechanical properties of Ihram Cotton Fabric
I have completed my review on manuscript membranes-1885924, entitled, “Theoretical and experimental study of the effect of plasma characteristics on the mechanical properties of Ihram Cotton Fabric.” The effects of plasma parameters on the mechanical properties of Ihram Cotton Fabric were investigated experimentally and theoretically using the radial distribution function of the electron temperature for argon glow discharge, the abnormal glow region in a low-density plasma, and weakly ionized gas for both configurations and .
Merit of this manuscript
Overall, the research topic is interesting and deserves to be published in membranes, but my comments and concerns must be addressed first.
Comments for authors regarding major revisions
Comment 1. Nonthermal plasma has many applications in a wide range of fields. When comparing nonthermal plasma applications, the authors' summary is insufficient to describe the significance of plasma. The background of nonthermal plasma should be expanded for new readers. I advise authors to include a diverse range of nonthermal plasma applications. I have some recommendations for authors to follow the studies and include them in the introduction section of this paper.
A) Scavenging effect of nonthermal plasma
- Ghimire, B., Lee, G. J., Mumtaz, S. & Choi, E. H. Scavenging effects of ascorbic acid and mannitol on hydroxyl radicals generated inside water by an atmospheric pressure plasma jet. AIP Adv. 8, (2018). [https://doi.org/10.1063/1.5037125].
- Zhang, L. et al. In-duct grating-like dielectric barrier discharge system for air disinfection. J. Hazard. Mater. 435, 129075 (2022). [https://doi.org/10.1016/j.jhazmat.2022.129075].
B) Application of plasma in agriculture for plant growth
- Lamichhane, P. et al. Low-Temperature Plasma-Assisted Nitrogen Fixation for Corn Plant Growth and Development. International Journal of Molecular Sciences vol. 22 (2021). [https://doi.org/10.3390/ijms22105360].
- Mildaziene, V., Ivankov, A., Sera, B. & Baniulis, D. Biochemical and Physiological Plant Processes Affected by Seed Treatment with Non-Thermal Plasma. Plants 11, (2022). [https://doi.org/10.3390/plants11070856].
- Domonkos, M., Tichá, P., Trejbal, J. & Demo, P. Applications of Cold Atmospheric Pressure Plasma Technology in Medicine, Agriculture and Food Industry. Applied Sciences 11, (2021). [https://doi.org/10.3390/app11114809].
C) Use of nonthermal plasma against brain tumors.
- Mumtaz, S., Rana, J. N., Choi, E. H. & Han, I. Microwave Radiation and the Brain: Mechanisms, Current Status, and Future Prospects. International Journal of Molecular Sciences vol. 23 (2022). [https://doi.org/10.3390/ijms23169288].
- Limanowski, R., Yan, D., Li, L. & Keidar, M. Preclinical Cold Atmospheric Plasma Cancer Treatment. Cancers 14, (2022). [https://doi.org/10.3390/cancers14143461].
Comment 2. The writing quality and English must need to improve. Some places authors indicate doubles spaces (line 52), paragraph starts with full stop (line 52), words different size (line 72 – 77), ??? at (line 103), ?? at (line 170), and line 320 – 324. These serious issues need to be fixed.
Comment 3. The novelty and scientific soundness of the work need to indicate clearly. For that the abstract need to be improved. Furthermore, at line 27, in abstract … interaction type, and gas type”… try to avoid repetition of same work “type.”
Comment 4. What is the meaning of word “Strengthen???” and highlighted as bold in line 103? It is not clear to me why this word is highlighted. Similarly, “the following readership??” in line 170.
Comment 5. Figure 3 need to be revised. Please remove the word "Chart title" from the figure and describe your conditions in a free space rather than on the graph.
Comment 6. Rewrite the conclusion as a single paragraph.
Comment 7. Write the and as an equation rather than writing R<L and R>L in overall text of this manuscript.
Comment 8: There are typos and inaccuracies in the paper, I recommend authors to read precisely and correct the grammatical errors.
Induct 12 3 4
1. Zhang, L. et al. In-duct grating-like dielectric barrier discharge system for air disinfection. J. Hazard. Mater. 435, 129075 (2022).
2. Mildaziene, V., Ivankov, A., Sera, B. & Baniulis, D. Biochemical and Physiological Plant Processes Affected by Seed Treatment with Non-Thermal Plasma. Plants 11, (2022).
3. Domonkos, M., Tichá, P., Trejbal, J. & Demo, P. Applications of Cold Atmospheric Pressure Plasma Technology in Medicine, Agriculture and Food Industry. Applied Sciences 11, (2021).
4. Limanowski, R., Yan, D., Li, L. & Keidar, M. Preclinical Cold Atmospheric Plasma Cancer Treatment. Cancers 14, (2022).
Author Response
answer of the comments

Round 2
Reviewer 2 Report
Review on the revised Theoretical and experimental study of the effect of plasma characteristics on the mechanical properties of Ihram Cotton Fabric
Comments for authors regarding major revisions
The authors must pay attention to all comments and carefully read and revise the manuscript accordingly.
The authors responses to comment 1, Comment 2, Comment 4, Comment 7, and Comment 8 and regarding modifications in the revised manuscript is not convincing to me. Furthermore, it is still not understandable to me what is the meaning of symbol “??” at page 6, line 174 and authors response to me is just “Done” which is very hard to figure out. Authors must carefully improve the background and add the articles mentioned in comment 1 and explain them in the introduction for background knowledge.
In response to previous comment 7, authors did some modifications in response to my comment but some part left as it is, from lines 218 – 325, it is again recommended to write the and as an equation rather than writing R<L and R>L. Also, cross check in overall text of this manuscript.
Also, the English writing in the revised version is still poor, with many grammatical errors that make it difficult to comprehend the precise meaning of statements. Some of them are listed here for your convenience:
- page 9, line 203 – 208), there is no full stop and “The” used as capital.
- Line 60, “Nonthermal plasma have gained a broad interest,” the have should be replaced by has and remove comma after interest.
- Line 61, due the ions should be revised as due to the ions.
- Line 63, application should be replaced by applications.
- Line 64, …and in the treatment… delete in.
- Line 68, … returned back to hajj and umrah… remove back.
- Line 75, … provide an excellent… remove an.
- Lines 125 – 127, this text was repeated in lines 132 – 133.
- Line 136, put space after full stop.
- Line 184, …results in… should be replaced by resulting in.
- Line 199, Substituting by equations, should be replaced as Substituting equations.
Previous major comments for authors which still needs to be resolved.
Comment 1. Nonthermal plasma has many applications in a wide range of fields. When comparing nonthermal plasma applications, the authors' summary is insufficient to describe the significance of plasma. The background of nonthermal plasma should be expanded for new readers. I advise authors to include a diverse range of nonthermal plasma applications. I have some recommendations for authors to follow the studies and include them in the introduction section of this paper.
A) Scavenging effect of nonthermal plasma
- Ghimire, B., Lee, G. J., Mumtaz, S. & Choi, E. H. Scavenging effects of ascorbic acid and mannitol on hydroxyl radicals generated inside water by an atmospheric pressure plasma jet. AIP Adv. 8, (2018). [https://doi.org/10.1063/1.5037125].
- Zhang, L. et al. In-duct grating-like dielectric barrier discharge system for air disinfection. J. Hazard. Mater. 435, 129075 (2022). [https://doi.org/10.1016/j.jhazmat.2022.129075].
B) Application of plasma in agriculture for plant growth
- Lamichhane, P. et al. Low-Temperature Plasma-Assisted Nitrogen Fixation for Corn Plant Growth and Development. International Journal of Molecular Sciences vol. 22 (2021). [https://doi.org/10.3390/ijms22105360].
- Mildaziene, V., Ivankov, A., Sera, B. & Baniulis, D. Biochemical and Physiological Plant Processes Affected by Seed Treatment with Non-Thermal Plasma. Plants 11, (2022). [https://doi.org/10.3390/plants11070856].
- Domonkos, M., Tichá, P., Trejbal, J. & Demo, P. Applications of Cold Atmospheric Pressure Plasma Technology in Medicine, Agriculture and Food Industry. Applied Sciences 11, (2021). [https://doi.org/10.3390/app11114809].
C) Use of nonthermal plasma against brain tumors.
- Mumtaz, S., Rana, J. N., Choi, E. H. & Han, I. Microwave Radiation and the Brain: Mechanisms, Current Status, and Future Prospects. International Journal of Molecular Sciences vol. 23 (2022). [https://doi.org/10.3390/ijms23169288].
- Limanowski, R., Yan, D., Li, L. & Keidar, M. Preclinical Cold Atmospheric Plasma Cancer Treatment. Cancers 14, (2022). [https://doi.org/10.3390/cancers14143461].
Comment 2. The writing quality and English must need to improve. Some places authors indicate doubles spaces (line 52), paragraph starts with full stop (line 52), words different size (line 72 – 77), ??? at (line 103), ?? at (line 170), and line 320 – 324. These serious issues need to be fixed.
Comment 4. What is the meaning of word “Strengthen???” and highlighted as bold in line 103? It is not clear to me why this word is highlighted. Similarly, “the following readership??” in line 170.
Comment 7. Write the and as an equation rather than writing R<L and R>L in overall text of this manuscript.
Comment 8: There are typos and inaccuracies in the paper, I recommend authors to read precisely and correct the grammatical errors.
Author Response
I answer about all the comments of reviewer 2 ( round 2)

Round 3
Reviewer 2 Report
Review on the revised Theoretical and experimental study of the effect of plasma characteristics on the mechanical properties of Ihram Cotton Fabric
In this draft, the authors addressed some of my comments and concerns. The quality of the revised version was enhanced by the authors. The current version of the manuscript is far superior to the old one. My advice is to accept the article.